# Nutritional Care of Hospitalized Children in Belgium: A Follow-Up Survey

**DOI:** 10.3390/nu17040718

**Published:** 2025-02-18

**Authors:** Marlies Destoop, Yvan Vandenplas, Marc Raes, Bruno Hauser, Elisabeth De Greef, Koen Huysentruyt

**Affiliations:** 1Department of Paediatric Gastro-Enterology, KidZ Health Castle, Universitair Ziekenhuis Brussel, Vrije Universiteit Brussel (VUB), 1090 Brussels, Belgium; marlies.destoop@vub.be (M.D.); yvan.vandenplas@uzbrussel.be (Y.V.); bruno.hauser@uzbrussel.be (B.H.); elisabeth.degreef@uzbrussel.be (E.D.G.); 2Department of Paediatrics, Jessa Ziekenhuis, 3500 Hasselt, Belgium; marc.raes@jessazh.be; 3Belgische Vereniging voor Kindergeneeskunde (BVK), 1050 Brussels, Belgium

**Keywords:** disease-associated malnutrition (DAM), pediatric malnutrition, nutritional screening, barriers to screening

## Abstract

**Background**: A 2014 survey showed nutritional management could be improved in Belgian pediatric departments. This follow-up survey aimed to: (1) list allied health resources/staffing in Belgian pediatric departments, (2) survey nutritional screening and follow-up, and (3) identify barriers. **Methods**: A nationwide survey (February–April 2021) via national and regional pediatric associations. **Results**: 61/90 (67.8%) of Belgian pediatric departments responded (80.1% of all Belgian pediatric hospital beds); 60.7% of the respondents were from larger centers (LCs, ≥20 beds). A dietitian was present in 80.3% of all responding units (LCs vs. smaller centers (SCs): *p* = 0.133), compared to 46.5% in the 2014 survey. Most dietitians seldom or never participate in ward rounds (86.9%) and participate only ad hoc to case discussions (72.1%). Systematic nutritional screening is implemented in 32.8% of pediatric departments. The screening tool STRONGkids is used in 30% of responding centers, compared to 21% in 2014. The most common barriers to conducting nutritional screening were lack of time (59.0%), a lack of knowledge (47.5%), and a lack of staff (42.6%). In French-speaking centers (FrCs), a positive screening result most often led to referral to a dietitian (86.7%), whereas in Dutch-speaking centers (DuCs), it more frequently resulted in a discussion with the pediatrician about nutritional management (54.3%) than referral to a dietitian (34.8%). Nutritional follow-up after discharge is most often conducted by a physician, with or without the involvement of a dietitian (95.1%), rather than a dietitian alone (3.3%). Malnutrition management barriers included “no barriers” (50.8%), a lack of knowledge (34.4%), a lack of reimbursement (24.6%), and a lack of time (24.6%). The barriers remain largely unchanged compared to 2014. **Conclusions**: The increasing availability of dietitians and the use of a screening tool in pediatric departments suggest an encouraging but limited improvement in nutritional care in Belgium. Persistent barriers that have remained unchanged since 2014 continue to hinder substantial advancements in nutritional care.

## 1. Introduction

Disease-associated malnutrition (DAM) in hospitalized children is associated with a higher complication rate, longer hospital stay, and consequently a higher cost [1,2,3,4,5]. The early recognition and treatment of malnutrition can improve the outcomes of children during illness. Therefore, the adequate evaluation of nutritional status and nutrition is an obvious step in the management of children during illness. Systematic nutritional screening is recommended as a strategy to identify children with (or at risk of) malnutrition in a recently published position paper by the Special Interest Group Clinical Malnutrition of the European Society for Paediatric Gastroenterology, Hepatology and Nutrition (ESPGHAN) [6].

We conducted a nationwide survey amongst Belgian pediatric hospitals in 2014, demonstrating an underestimation of the prevalence of malnutrition by pediatricians and a lack of awareness and knowledge regarding the nutritional screening process [7]. Almost a decade after this first study, a follow-up study was performed.

The purpose of this study was to (1) list the currently available dietetic resources and staffing in Belgian pediatric departments; (2) survey current practices for nutritional screening, assessment, management, and follow-up; and (3) identify possible barriers to adequate nutritional care and follow-up.

## 2. Materials and Methods

Data collection: A cross-sectional questionnaire study was conducted to collect data on nutritional screening and clinical practice for nutritional assessment and management in the pediatric wards of Belgian hospitals. A first draft of the questionnaire was discussed by a group of experts. Belgium has 3 official languages. Dutch and French are the most widely spoken, while German, though an official language, is spoken by a very small population. Consequently, hospitals in Belgium operate with either Dutch or French as their primary language. Therefore, the questionnaire was only made available in Dutch or French. The questionnaire was originally developed in Dutch and later translated into French. Next, the French version was translated back into Dutch, making it possible to compare both versions and ensure that the meaning was accurately preserved across languages. The English version of the questionnaire is available in Appendix A.

The 90 department heads of all Belgian pediatric departments of secondary and tertiary-level hospitals were invited to participate in the survey via email and via postal mail in February 2021. A reminder was sent, through the post and by email, 2 months later. Participation in the study was on a voluntary basis. Respondents could opt out of the questionnaire at any time and were not compensated for their contribution. Data collecting was conducted between February 2021 and April 2021. The approval of the ethics committee (Medical Ethics Committee UZ Brussel) was obtained prior to commencing the study. Since language differences may be associated with cultural differences, we regarded language as an important factor. Dutch is the dominant language in the north of the country (which is called the Flemish region), while French dominates the south (which is called the Walloon region). Participating centers were divided into French-speaking and Dutch-speaking, based on the geographical origin. Hospitals based in Brussels were divided into French-speaking and Dutch-speaking based on the primary language used in the hospital.

Statistical analysis: Statistical analysis was performed using the statistical software IBM SPSS Statistics 16. A chi-square or Fisher’s exact test was used to analyze differences in proportion. Continuous variables were compared using Student’s *t*-test or a Mann–Whitney U test where appropriate. A *p*-value of <0.05 was considered significant.

## 3. Results

### 3.1. Respondent Characteristics

In total, 61 out of 90 (67.8%) Belgian pediatric departments responded, representing 80.1% of all pediatric hospital beds in Belgium. Of the participating departments, 46 (75.4%) were Dutch-speaking, covering 76.0% of all Dutch-speaking pediatric hospital beds, and 15 were French-speaking, covering 87.0% of French-speaking pediatric hospital beds. Overall, 60.7% of the respondents were from larger centers (LCs; ≥20 beds). There was no significant difference in the proportion of respondents from smaller centers (SC; <20 beds) between Dutch- and French-speaking departments (*p* = 0.363). Table 1 provides the demographic characteristics of the study population.

### 3.2. Perceived Prevalence of DAM and Organizational Aspects

All respondents acknowledged that malnourished children are admitted to their centers, albeit at varying frequencies. The perceived prevalence of malnutrition was significantly different between the Dutch (DuC)- and French-speaking centers (FrCs) (*p* ≤ 0.001): 40.0% of the FrCs reported admitting a malnourished child ≥1x/week, in comparison to 2.2% of the DuCs. In total, 32.6% of the DuCs and 26.7% of the FrCs reported admitting a malnourished child between 1x/week and 1x/month. Most of the DuCs (65.2%) reported admitting malnourished children less than once per month, compared to 33.3% of the FrCs.

A dietitian was present in 80.3% of all responding units with no significant difference between LCs and SCs. However, there was a significant difference in the number of full-time dietitians: 32.4% of the LCs had more than one full-time dietitian, compared to only 4.2% of the SCs (*p* = 0.016). The clinical responsibilities of dietitians differed significantly between LCs and SCs (Figure 1). Although the systematic participation of dietitians in ward rounds was rare (8.2%), they were frequently involved on an ad hoc basis (72.1%). Nutritional interventions such as supplementary feeding and tube feeding generally fell under the dietitians’ responsibilities in larger centers, whereas this was less common in smaller centers. The initiation and follow-up of parenteral nutrition was not within their responsibilities (Figure 1).

### 3.3. Screening

Systematic nutritional screening was conducted in 32.8% of the participating hospitals, with no significant difference between SCs and LCs, or between FrCs and DuCs. The majority of the hospitals (59.0%) reported only screening in the case of a clinical suspicion of malnutrition or feeding difficulties. A small percentage of hospitals screened only occasionally (6.6%), while 2.7% reported never screening (Figure 2). The majority (67.2%) did not use any screening tools. When a screening tool was used, STRONGkids was the most commonly applied, being used in 29.5% of the participating centers. In FrCs, a positive screening result most often led to referral to a dietitian (86.7%), whereas in DuCs it more frequently resulted in a discussion with the pediatrician about nutritional management (54.3%) or referral to a dietitian (34.8%).

### 3.4. Nutritional Assessment

When asked about clinical practices regarding the assessment of malnutrition, 77.0% of the respondents reported always using weight and height plotted on a growth curve or calculated as a z-score. Only 26.2% of the centers reported using a protocol for nutritional assessment, and, of these, 14.8% followed the protocol published by the Flemish Pediatric Society (“Vereniging voor Kindergeneeskunde”) [8]. When asked about this protocol during the survey, most centers (72.1%) indicated they did not currently use it but expressed interest in implementing it. There was no significant difference between SCs and LCs.

All participating centers indicated that they use age-appropriate weighing scales. For measuring height and length, 86.9% of the centers reported using a measuring board for children under 1 m, 65.6% use a stadiometer, 9.8% use a tape measure alongside the child, and only 1.6% use segmental measurements. No significant difference was found in the methods of measuring children between LCs and SCs.

For diagnosing malnutrition, 98.4% of the centers reported using weight, height, BMI percentiles, and/or z-scores. Notably, half of the participating centers reported using blood or serum markers for assessing macronutrient levels, despite this practice not being supported by current guidelines [6]. This practice was significantly more frequent in FrCs at 86.7%, compared to 43.5% in DuCs (*p* = 0.006). A total of 41.0% of the participants reported using blood/serum markers for micronutrients, with no significant difference between DuCs and FrCs nor between SCs and LCs. A significant difference was found in the use of anthropometric measurements for body composition between DuCs (8.7%) and FrCs (33.3%) during nutritional assessments (*p* = 0.033) (Figure 3).

### 3.5. Management

Only 19.7% of the hospitals have a protocol in place for the treatment of malnutrition, with 29.7% of LCs and 4.2% of SCs having such a protocol (*p* = 0.049). There was no significant difference in the use of a treatment protocol for malnutrition between DuCs and FrCs. The majority of the respondents (60.7%) reported that their hospital tracks the intake of admitted children by estimating how much of each meal was consumed. Only 3.3% count the calories in the food, 4.9% conduct mealtime audits, and 31.1% do not monitor intake routinely. For children at risk of malnutrition, the monitoring of intake is slightly different: half of the participants indicated that they monitor intake by estimating meal consumption, 19.7% count the calories in the food, 14.8% use mealtime audits, 8.2% employ all three methods, and only 4.9% do not routinely monitor intake for these children (Table 2).

### 3.6. Follow-Up

Only 26.7% of the hospitals indicated that they always provide information on nutritional status and management in the discharge letter. In the majority (52.5%), nutritional status and management are included in the discharge letter only if a nutritional intervention was performed, such as supplementary feeding or the provision of parenteral nutrition. Nutritional follow-up after discharge is most often conducted by a physician, with or without the involvement of a dietitian (95.1%), rather than a dietitian alone (3.3%). No significant difference was found between SCs and LCs regarding the follow-up.

### 3.7. Barriers

The most common barriers to conducting nutritional screening were a lack of time (59.0%), a lack of knowledge (47.5%), and a lack of staff (42.6%). No significant difference was found between SCs and LCs regarding barriers for screening. For assessing the nutritional status, the primary barriers were also a lack of time (59.0%) and a lack of knowledge (50.0% in SCs, 21.6% in LCs; *p* = 0.022). The most frequent barriers to treating malnutrition included “no barriers” (50.8%), a lack of knowledge (34.4%), a lack of reimbursement (24.6%), and a lack of time (24.6%). No significant difference was found between SCs and LCs regarding barriers for management (Figure 4).

### 3.8. Training

There was a lot of interest in a workshop on optimizing nutritional care (91.8% of participants). A total of 83.6% expressed interest in support for developing and implementing a protocol and 82.0% in a clinical training center.

## 4. Discussion

This nationwide study was conducted to follow up on a similar survey study from 2014 by our group, which examined the clinical practices regarding nutritional screening in Belgian pediatric departments [7]. As a result of this previous study, it became clear that there was a lack of awareness and knowledge regarding pediatric nutritional care. At that time, no pediatric guidelines for malnutrition screening were available. In 2016, we published an algorithm for screening undernutrition in hospitalized children, in direct response to the 2014 survey [8]. We aimed to see if practices had changed since that initial study.

In 2014, nutritional screening was not systematically used in pediatric departments in Belgium, with fewer than one in four DuCs screening systematically. Today, systematic nutritional screening has slightly improved in DuCs, with almost one in three centers now conducting systematic screening. However, FrCs appear to have reduced their screening practices, with 4 out of 10 departments currently screening systematically compared to 6 out of 10 in 2014. This apparent decline may reflect an evolving understanding of what constitutes “screening”. Over the years, there has been a growing emphasis on the use of screening tools, and what was considered screening in 2014 (measuring a child’s nutritional status) may now be understood as using specific screening tools. This is supported by the increased use of screening tools; in 2014, 21% of department heads reported using the STRONGkids screening tool, compared to 30% in 2021. While these findings suggest a gradual improvement in the adoption of screening tools, overall progress in systematic nutritional screening remains limited. A concerning 57% of hospitals only screen for malnutrition in case of clinical suspicion. This stagnation in clinical practice regarding screening is not unique to Belgium, similar patterns have been observed in other countries such as the USA and Canada [6,9]. These findings highlight the need for targeted strategies to promote the consistent implementation of nutritional screening in pediatric departments.

In contrast to a recent survey conducted in the USA, which reported a decline in dietitian staffing, we observed an encouraging improvement in the availability of dietitians in pediatric departments, with 80% of responding departments now having a dietitian compared to 47% in 2014 [9]. While this progress is notable, it should be interpreted cautiously. Availability alone does not equate to active involvement in patient care. Our study highlights that dietitians’ participation in patient management remains very limited, particularly in terms of regular engagement in ward rounds and interdisciplinary discussions with physicians. The importance of this is supported by a recent study conducted by Belanger et al. demonstrating the impact of nutritional counseling by dietitians on the evolution of body weight during hospitalization [10]. Recent studies demonstrated that 16–21% of hospitalized children lose weight during their hospital admission [11,12].

Awareness of malnutrition appears to have slightly improved since 2014. At that time, only 2.8% of respondents reported admitting a malnourished child at least once per week, compared to 11.5% in 2021. This number aligns more closely with the reported prevalence in the Belgian population [13]. Although, in the literature worldwide, the prevalence is highly variable, ranging between 2% and 50%, depending on assessment methods, malnutrition definitions, and the study population (which could be disease-specific or geographically specific) [1,14,15,16,17,18,19,20,21,22].

Another area of concern is the assessment of malnutrition. One-third of responding centers use visual inspection to evaluate malnutrition, while only 1 in 5 large centers and 1 in 20 small centers use anthropometric measurements to assess body composition. Additionally, only one in four centers report using a standardized protocol for nutritional assessment, highlighting significant room for improvement in this area.

The limited progress in clinical practice for nutritional care since 2014 may stem from the barriers to nutritional screening, assessment, and management, which have remained largely unchanged since the previous study. These persistent barriers continue to prevent significant improvements, underscoring the need to identify and address them.

A lack of knowledge continues to be a major barrier to all three aspects of nutritional care (screening, assessment, and management) regarding malnutrition. This challenge seems to be universal as similar barriers have been reported in Canada [23,24]. However, the participating centers are eager to improve, as evidenced by their interest in workshops and support for developing and implementing a protocol. More than 70% of the centers expressed interest in adopting the protocol approved by the Flemish Pediatric Society (VVK), and nearly 15% are already using it. Organizing workshops and providing support for protocol implementation could help address these barriers.

Other barriers include a lack of staff and time, which are also reported in other parts of the world such as the USA and Canada [9,25]. To overcome these, screening should be made as simple, automatic, and time-efficient as possible. The STRONGkids tool, recommended in the screening algorithm, is convenient because it does not require anthropometric measurements [8,26]. However, it is important to recognize that implementing a screening tool does not necessarily improve staff awareness or patient outcomes, as shown in a study by Marderfeld [27]. It cannot replace good training of the staff to increase their awareness of the importance of screening for and assessment and treatment of malnutrition. The use of multidisciplinary unit-based champion teams has been shown to lead to an increase in the identification of patients at nutritional risk in the USA and a training program by a nutrition support team in France led to an improvement in overall nutritional knowledge amongst staff members and an improvement in the frequency of obtaining anthropometric measurements [28,29]. Nutrition-focused quality improvement programs, focusing on risk screening, assessment, and intervention in pediatric care, are also being explored. This approach has proven successful in benchmarking nutritional care for adult patients [30].

Finally, the lack of reimbursement for supplementary feeding is considered an important barrier for the management of malnutrition. This issue should be addressed with the Belgian government.

The response rate in this study was 67.8%, which is comparable to the response rate of 73.2% in our previous study in 2014. The comparable response rates between the two studies suggest consistency in the engagement of pediatric departments over time. While a higher response rate is always desirable to reduce the risk of non-response bias, we believe that the current response rate provides a robust representation of the target population. Future studies could explore additional strategies, such as additional reminders, to further optimize response rates.

When comparing the 2014 and 2021 studies, it is important to note that tertiary-level hospitals were not included in 2014. This difference in study populations may affect the comparability of the findings between the two studies. However, aside from this, the study populations are generally similar, with a slight overrepresentation of Dutch-speaking respondents in the current study. This overrepresentation could introduce a potential bias, particularly in aspects where language or regional differences may play a role. However, French-speaking hospitals are well represented when considering the number of pediatric beds.

Another limitation of this study is that many respondents completed the questionnaire on paper, which allowed for some questions to be skipped. Fortunately, this occurred in only a few cases: one respondent skipped one question, another skipped three questions, and a third skipped four questions. These were considered missing answers. For any given question, at most one respondent skipped their answer, so we do not believe this introduced significant bias. Additionally, two respondents selected multiple answers where only one option was allowed (regarding the guidelines for weighing hospitalized children in their center). These contradictory answers were treated as skipped responses, and this question was ultimately excluded from the manuscript. Future studies may benefit from using digital surveys to minimize these issues and improve data completeness.

## 5. Conclusions

Our research highlights encouraging still limited progress in nutritional care practices in Belgian pediatric departments, potentially reflecting a growing awareness of malnutrition. However, significant areas for improvement remain such as the involvement of dietitians in clinical practice and the use of standardized protocols for screening, nutritional assessment, and management.

Our findings suggest that staff in Belgian pediatric departments are motivated to implement positive changes, but numerous barriers hinder progress. Notably, these barriers have remained largely unchanged over the last decade, underscoring the need to address them and develop practical solutions. One potential approach could be to assist pediatric centers in implementing a simple, time-efficient algorithm for the screening, assessment, and management of malnutrition. Additionally, organizing workshops to enhance staff knowledge and supporting the adoption of standardized protocols could help drive meaningful and sustainable improvements in nutritional care.

## Figures and Tables

**Figure 1 nutrients-17-00718-f001:**
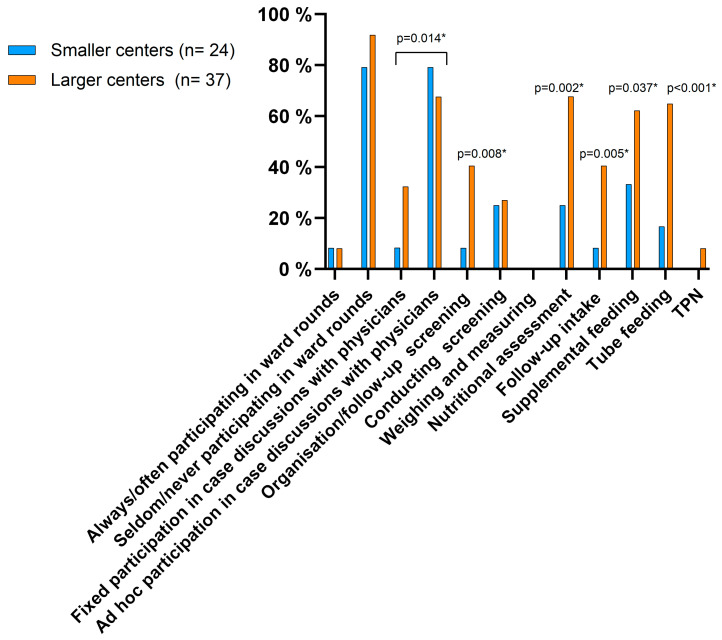
The responsibilities of the dietitians in pediatric centers, as reported in the survey, categorized by center size: smaller centers (<20 beds) and larger centers (≥20 beds). TPN, total parental nutrition; * difference between smaller (<20 beds) and larger (≥20 beds) pediatric centers.

**Figure 2 nutrients-17-00718-f002:**
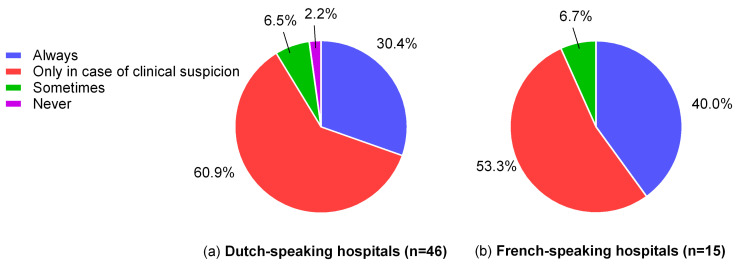
The use of systematic nutritional screening in pediatric centers as reported in the survey. (**a**) The use of systematic nutritional screening in Dutch-speaking centers; (**b**) the use of systematic nutritional screening in French-speaking centers (no significant differences).

**Figure 3 nutrients-17-00718-f003:**
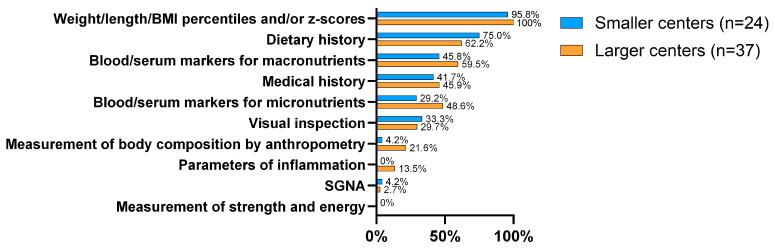
Methods used for the evaluation of malnutrition in pediatric centers, categorized by center size (no significant differences).

**Figure 4 nutrients-17-00718-f004:**
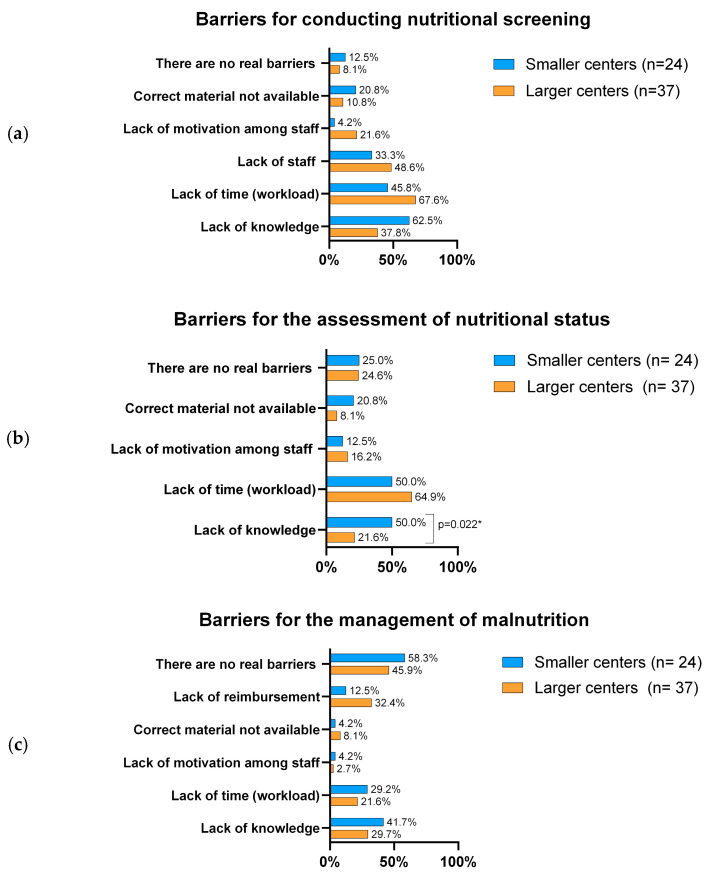
Barriers reported by pediatric centers in relation to nutritional care, categorized by center size. (**a**) Barriers for conducting nutritional screening; (**b**) barriers for measuring nutritional status; (**c**) barriers for the management of malnutrition. * The difference between smaller centers and larger centers.

**Table 1 nutrients-17-00718-t001:** Demographic characteristics of the study population.

	Total (N = 61) N (%)	DuC (N = 46) N (%)	FrC (N = 15) N (%)	*p*-Value *
Type of hospital				
Tertiary level	10 (16.4%)	4 (8.7%)	6 (40.0%)	0.010
Secondary level	51 (83.6%)	42 (91.3%)	9 (60.0%)	
Type of medical record				
Electronic	57 (93.4%)	43 (93.5%)	14 (93.3%)	1.000
Combination of electronic and paper	4 (6.6%)	3 (6.5%)	1 (6.7%)	
Mode of response				
Email	25 (41.0%)	21 (45.7%)	4 (26.7%)	0.238
Post	36 (59.0%)	25 (54.3%)	11 (73.3%)	
Number of beds				
<20 beds	24 (39.3%)	20 (43.5%)	4 (26.7%)	0.363
≥20 beds	37 (60.7%)	26 (56.5%)	11 (73.3%)	

* Difference between Dutch-speaking centers (DuCs) and French-speaking centers (FrCs).

**Table 2 nutrients-17-00718-t002:** Management of malnutrition.

	Total (N = 61) N (%)	SC (N = 24) N (%)	LC (N = 37) N (%)	*p*-Value *
Protocol for treatment of malnutrition				0.049
- Yes	12 (19.7%)	1 (4.2%)	11 (29.7%)	
- No, but it is under development	11 (18.0%)	5 (20.8%)	6 (16.2%)	
- No	38 (62.3%)	18 (75.0%)	20 (54.1%)	
Monitoring intake of admitted children				0.511
- Yes, through tracking/counting of calories in the food	2 (3.3%)	0 (0.0%)	2 (5.4%)	
- Yes, by estimating how much of the meal was eaten	37 (60.7%)	15 (62.5%)	22 (59.5%)	
- Yes, via mealtime audits	3 (4.9%)	2 (8.3%)	1 (2.7%)	
- No, intake is not routinely monitored for all children	19 (31.1%)	7 (29.2%)	12 (32.4%)	
Monitoring intake of children at risk of malnutrition				0.054
- Yes, through tracking/counting of calories in the food	12 (19.7%)	3 (12.5%)	9 (24.3%)	
- Yes, by estimating how much of the meal was eaten	32 (52.5%)	11 (45.8%)	21 (56.8%)	
- Yes, via mealtime audits	9 (14.8%)	6 (25.0%)	3 (8.1%)	
- Yes, via the 3 methods above	5 (8.2%)	1 (4.2%)	4 (10.8%)	
- No, intake is not routinely monitored for all children	3 (4.9%)	3 (12.5%)	0 (0.0%)	

* Difference between smaller centers (SCs) (<20 beds) and larger centers (LCs) (≥20 beds).

## Data Availability

The original contributions presented in this study are included in the article/Appendix A. Further inquiries can be directed to the corresponding author.

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
