# Peer review of "Nutritional Care of Hospitalized Children in Belgium: A Follow-Up Survey"

_nutrients, 2025, doi:10.3390/nu17040718_

Round 1
Reviewer 1 Report
Comments and Suggestions for Authors
The authors present an interesting study that acts as a follow on of sorts to a previous study of theirs in which the nutritional care afforded to hospitalised children in Belgian healthcare facilities was examined. Briefly , the authors aimed to examine what roles nutritionists and other healthcare providers played in terms of managing nutrition in in taken children patients, and whether routine screening was performed to determine the optimal diet for those who may be in recovery while in the facilities care. Aspects of this are quantified across French and Dutch speaking facilities, with comparisons drawn at this level and whether the facility was regarded large scale or otherwise. Overall, the data suggests that the situation is largely unchanged with many of the same barriers still remaining and preventing any improvement on the nutritional aspect of care in this region.
In reviewing the manuscript I made a couple of observations. The following should be considered by the authors when preparing a suitable revision.
1. While the writing of the manuscript is very good for the most part, there are some instances of typos/errors in grammar that need to be corrected. For example, in Figure 1 I would suggest labelling the diagram ‘Responsibilities of Dieticians’ rather than ‘Responsibility Dieticians’. The authors should review the entire manuscript and correct for errors such as the above in any future resubmission.
2. It is advised that the authors include the n-number in figures such as Figure 1 and 2 as a reminder that the comparisons being made are not equal in n-number.
3. The labelling in Figure 2 needs to be improved – what is the question that is being surveyed? It is inferred via the main body of text, but the figure itself should present that the answers refer to in a clearer manner. Also, could this data be examined from a statistical sense to compare Dutch and French speaking institutes?
4. In Figure 3, were any statistical tests performed on the data?
5. Why are data sets located in the appendix and not the main text? Given that the data is spoken about at length it would make more sense to me to include it in the main article. The authors should give reasoning for why this data was relocated to an appendix and consider whether this is the best place for it.
6. The discussion does a good job of relating the findings in this study to other data from other articles, however it would be good if the authors were able to include more studies that support the findings/suggestions/recommendations made in this study. I felt this was somewhat light overall, and feel more could evidence could be referred to in order to support the points made.
7. While it is appreciated that the authors included limitations in their discussion I feel this should be fleshed out more. More limitations should be highlighted with respect to this current study, and more information on the point that many surveys were incomplete should be given.
Author Response
|
Reviewer 1 |
|
|
The authors present an interesting study that acts as a follow on of sorts to a previous study of theirs in which the nutritional care afforded to hospitalised children in Belgian healthcare facilities was examined. Briefly , the authors aimed to examine what roles nutritionists and other healthcare providers played in terms of managing nutrition in in taken children patients, and whether routine screening was performed to determine the optimal diet for those who may be in recovery while in the facilities care. Aspects of this are quantified across French and Dutch speaking facilities, with comparisons drawn at this level and whether the facility was regarded large scale or otherwise. Overall, the data suggests that the situation is largely unchanged with many of the same barriers still remaining and preventing any improvement on the nutritional aspect of care in this region.
|
We thank the reviewer for this comment. |
|
We have improved the quality and readability of our figures. |
|
|
In reviewing the manuscript I made a couple of observations. The following should be considered by the authors when preparing a suitable revision. While the writing of the manuscript is very good for the most part, there are some instances of typos/errors in grammar that need to be corrected. For example, in Figure 1 I would suggest labelling the diagram ‘Responsibilities of Dieticians’ rather than ‘Responsibility Dieticians’. The authors should review the entire manuscript and correct for errors such as the above in any future resubmission.
|
Thank you for your thorough review. We acknowledge the typo in Figure 1 and have corrected the label to read ‘Responsibilities of Dieticians’ as suggested. Additionally, we have conducted a detailed review of the manuscript to identify and correct any other typos or grammatical errors. |
|
It is advised that the authors include the n-number in figures such as Figure 1 and 2 as a reminder that the comparisons being made are not equal in n-number.
|
We agree that including the n-numbers in Figures 1 and 2 would provide important context for the comparisons being made. We have updated all figures to include the n-numbers for clarity and to remind readers of the differences in sample sizes. |
|
The labelling in Figure 2 needs to be improved – what is the question that is being surveyed? It is inferred via the main body of text, but the figure itself should present that the answers refer to in a clearer manner. Also, could this data be examined from a statistical sense to compare Dutch and French speaking institutes?
|
We agree that the labelling in Figure 2 should be improved for clarity. In response, we have revised both the figure label and the answer options to ensure that the question being surveyed is clearly presented within the figure itself.
Additionally, we have examined the data for statistical differences between Dutch- and French-speaking centers. As noted in the manuscript, there was no significant difference observed: "Systematic nutritional screening was conducted in 32.8% of the participating hospitals, with no significant difference between SC and LC, or between FrC and DuC." (also added to the Figure) |
|
In Figure 3, were any statistical tests performed on the data?
|
Yes, statistical tests were performed on the data in Figure 3 to compare smaller centers and larger centers. As noted in the manuscript, no statistically significant differences were found between these groups. (also added to the Figure) |
|
Why are data sets located in the appendix and not the main text? Given that the data is spoken about at length it would make more sense to me to include it in the main article. The authors should give reasoning for why this data was relocated to an appendix and consider whether this is the best place for it.
|
We initially placed the data sets in the appendix to maintain the flow and readability of the main text, as we wanted to avoid overloading the article with detailed tables. However, we agree with your suggestion to move the data sets to the main text. We have updated the manuscript accordingly and integrated the data sets into the main text. |
|
The discussion does a good job of relating the findings in this study to other data from other articles, however it would be good if the authors were able to include more studies that support the findings/suggestions/recommendations made in this study. I felt this was somewhat light overall, and feel more could evidence could be referred to in order to support the points made. |
We appreciate your suggestion to include additional studies to support our findings and recommendations. In response, we have carefully reviewed the literature and incorporated more relevant studies to strengthen our discussion and provide additional evidence for our conclusions. |
|
While it is appreciated that the authors included limitations in their discussion I feel this should be fleshed out more. More limitations should be highlighted with respect to this current study, and more information on the point that many surveys were incomplete should be given. |
We appreciate the reviewer’s suggestion to expand the discussion of the study's limitations. In response, we have elaborated on the limitations section in the discussion. Specifically, we have now included a detailed account of the response rate (67.8%), comparing it to our previous study (73.2%), and discussing its implications for the generalizability of the findings. Additionally, we have clarified the issue of incomplete surveys. While the majority of respondents completed the questionnaire fully, we noted that three respondents skipped one or more questions, and two respondents selected multiple answers for a question where only one response was permitted. These instances were minimal and unlikely to introduce significant bias, but they were acknowledged as potential limitations. We also noted that future studies could benefit from using digital survey tools to minimize such issues and improve data completeness. We hope these additions adequately address the reviewer’s concerns and provide a more comprehensive discussion of the study’s limitations. |
Reviewer 2 Report
Comments and Suggestions for Authors
This study is based on a nationwide survey regarding pediatric nutritional management in Belgium. As a high-income, highly developed economy with universal health care coverage and ranking within the top 20 globally for GDP/capita, the medical service set up a standard for the world. The level of dietitians’ involvement in this study is very concerning, especially since most of them only provide ad hoc (as needed) services. It is shocking that one-third of the malnutrition evaluation uses visual inspection. The modest response rate (67.8%) is quite acceptable. The pediatric departments are arbitrarily classified as larger (>20 beds) or smaller. I am unsure how this classification compares to the level 1-4 system. The authors claim this is an update to the 2014 survey. This survey creates a good opportunity to investigate what has changed over the past 10 years. Unfortunately, the only change mentioned in the abstract is the increase in the dietitian’s presence during clinical care. Whether this increase is significant is not described.
The abstract describes how referrals to dietitians are more common in French-speaking centers than in Dutch-speaking centers. Are the two different centers comparable in their level of service or the size of their pediatric units? Does the comparison reach statistical significance?
I do not understand “(Lines 55-56) The questionnaire was set up in Dutch and later translated in French and back translated to Dutch.” Why not just use the original Dutch version instead of the Dutch version translated back from French?
Content in lines 65-74 belongs to the Data Collection instead of the statistical analysis.
I believe the content in lines 80-82 is from the original Children’s template, which should be removed when you transfer your manuscript to the template.
Since German-speaking is not included on purpose, the 90 units used to calculate the response rate will be wrong. Similarly, some calculations in section 3.1, except Table 1, should be revised with German-speaking centers removed.
The abbreviation cfr. is used in five places. I don't understand its meaning. Does it stand for “Code of Federal Regulations”, “Case Fatality Rate”, Italian “confronta”, or “compare”? My web search provided me with several possibilities without anyone suitable to put into the context.
Figure legend for Figure 1. Please explain what is “bilan” in the figure.
I do not concur with the conclusion that “nutritional awareness has improved in Belgian paediatric departments,” as the data clearly show a lack of dietitian involvement in pediatric practice (Figure 1) and that adequate anthropometric evaluation is done in less than 22% of patients suspected of malnutrition. This is also reflected in the second sentence of the conclusion. However, I agree entirely that the Pediatric Society needs to advocate the dietitians’ involvement and lobby the change to the federal policymakers and lawmakers.
I suggest changing some words in the manuscript.
- Dietitian instead of dietician. Dietician is an older spelling that is rarely used after 1930. Both spellings are used in the abstract and main text. Can you stick to one?
- Allied health instead of paramedics. Paramedics are used by professionals who manage patients emergently, usually outside hospitals.
- Line 40: change “with (risk for)” to “with (at risk of)”
- Line 105-106: “None of the respondents reported never admitting malnourished children.” This is a double-negative description and is confusing.
Author Response
|
Reviewer 2 |
|
|
This study is based on a nationwide survey regarding pediatric nutritional management in Belgium. As a high-income, highly developed economy with universal health care coverage and ranking within the top 20 globally for GDP/capita, the medical service set up a standard for the world. The level of dietitians’ involvement in this study is very concerning, especially since most of them only provide ad hoc (as needed) services. It is shocking that one-third of the malnutrition evaluation uses visual inspection. The modest response rate (67.8%) is quite acceptable. The pediatric departments are arbitrarily classified as larger (>20 beds) or smaller. I am unsure how this classification compares to the level 1-4 system. The authors claim this is an update to the 2014 survey. This survey creates a good opportunity to investigate what has changed over the past 10 years. Unfortunately, the only change mentioned in the abstract is the increase in the dietitian’s presence during clinical care. Whether this increase is significant is not described.
|
Thank you for your thorough and insightful review. We fully agree that the level of dietitians' involvement and the reliance on visual inspection for malnutrition evaluation by one-third of respondents are concerning findings. In response, we have emphasized these points more strongly in the discussion section to highlight their significance and implications. Additionally, as per your suggestion, we have expanded more detailed results regarding changes since the 2014 survey in the abstract. |
|
We have improved the quality and readability of our figures. |
|
|
The abstract describes how referrals to dietitians are more common in French-speaking centers than in Dutch-speaking centers. Are the two different centers comparable in their level of service or the size of their pediatric units? Does the comparison reach statistical significance?
|
Yes, the participating Dutch- and French-speaking centers are comparable in terms of the size of their pediatric units. As noted in Table 1, the comparison does not reach statistical significance. |
|
I do not understand “(Lines 55-56) The questionnaire was set up in Dutch and later translated in French and back translated to Dutch.” Why not just use the original Dutch version instead of the Dutch version translated back from French?
|
We understand that the phrasing may not have been entirely clear. The process of translating the questionnaire from Dutch to French and then back-translating it into Dutch was conducted to ensure accuracy and consistency between the two language versions. This method helps identify and resolve any potential discrepancies in translation, ensuring that both versions convey the same meaning.
We agree that this could be explained more clearly, and we have revised this in the manuscript to better articulate the rationale behind this approach. |
|
Content in lines 65-74 belongs to the Data Collection instead of the statistical analysis.
|
Agree, we have replaced this part to data collection. |
|
I believe the content in lines 80-82 is from the original Children’s template, which should be removed when you transfer your manuscript to the template.
|
You are correct that the content in lines 80-82 was from the template. We have removed this section. |
|
Since German-speaking is not included on purpose, the 90 units used to calculate the response rate will be wrong. Similarly, some calculations in section 3.1, except Table 1, should be revised with German-speaking centers removed.
|
You are correct in pointing out the need for clarification regarding the German-speaking population in Belgium. While German is indeed an official language in Belgium, it is only spoken by a very small population in Belgium (< 50,000 people). The German-speaking population generally speaks French or Dutch as a second language. Since this population is so small, no hospital in Belgium operates exclusively or primarily in German; all hospitals are either primarily Dutch- or French-speaking. We will clarify this point in the manuscript to avoid any misunderstandings. |
|
The abbreviation cfr. is used in five places. I don't understand its meaning. Does it stand for “Code of Federal Regulations”, “Case Fatality Rate”, Italian “confronta”, or “compare”? My web search provided me with several possibilities without anyone suitable to put into the context.
|
Cfr. is an abbreviation of the Latin term confer, which means "refer to." However, we decided to delete it in our text. |
|
Figure legend for Figure 1. Please explain what is “bilan” in the figure.
|
We agree that "bilan" is not an appropriate term in English, and we have replaced it with "assessment". |
|
I do not concur with the conclusion that “nutritional awareness has improved in Belgian paediatric departments,” as the data clearly show a lack of dietitian involvement in pediatric practice (Figure 1) and that adequate anthropometric evaluation is done in less than 22% of patients suspected of malnutrition. This is also reflected in the second sentence of the conclusion. However, I agree entirely that the Pediatric Society needs to advocate the dietitians’ involvement and lobby the change to the federal policymakers and lawmakers.
|
We have revised the conclusion so that it is more nuanced. We now emphasize that while there has been some modest improvement in awareness and clinical practice regarding malnutrition, significant challenges remain. |
|
I suggest changing some words in the manuscript: - Dietitian instead of dietician. Dietician is an older spelling that is rarely used after 1930. Both spellings are used in the abstract and main text. Can you stick to one?
|
We agree with your recommendation and have changed all instances of "dietician" to "dietitian" throughout the manuscript, as you proposed. |
|
Allied health instead of paramedics. Paramedics are used by professionals who manage patients emergently, usually outside hospitals.
|
We agree with your suggestion and have replaced "paramedics" with "allied health" throughout the manuscript. |
|
Line 40: change “with (risk for)” to “with (at risk of)
|
Corrected to “Systematic nutritional screening is recommended as a strategy to identify children with (or at risk of) malnutrition in a recently published position paper by the Special Interest Group Clinical Malnutrition of the European Society for Pediatric Gastroenterology, Hepatology and Nutrition (ESPGHAN) [6]” |
|
Line 105-106: “None of the respondents reported never admitting malnourished children.” This is a double-negative description and is confusing.
|
This is indeed confusing. We corrected it: “All respondents acknowledged that malnourished children are admitted to their centers, albeit at varying frequencies.” |
Round 2
Reviewer 1 Report
Comments and Suggestions for Authors
The authors have suitably addressed my comments.
Author Response
No comments from the reviewer
Reviewer 2 Report
Comments and Suggestions for Authors
You have made changes according to my comments, but unfortunately, not all of them as you claimed in your correspondence. One of them is the spelling of dietician can be seen in several places (lines 16, 18, 23, 25, 26, 27, 29, 106, 108, 109, 110, 112, 121, 133, 135, 190, 239, 240, and 246). In fact, you have changed all “dietitian” into “dietician”. I am also concerned about the conclusion in the abstract section as the second sentence contradicts the first sentence.
It is strange to me that In Figure 1, the p-value for organization/follow-up screening is 0.08, while it is 0.005 for follow-up intake, even with a similar distribution. Make sure the statistics are correct. It is well described in Figure 1 that small centers (<20 beds) and large centers (> 20 beds), so the same message does not need to be reiterated in Figures 3 and 4.
Do you mean “asked” instead of “informed” in line 146?
This is a follow-up study after your 2016 publication. I would like to know whether your pediatric society/organization launched a national initiative to advocate for the implementation of your proposed guidelines. Publishing a paper cannot guarantee that all readers will accept the conclusion unless there is an authoritative instruction.
Comments on the Quality of English LanguageI still have some difficulty understanding some of the sentences due to word choice or the way of expression.
Author Response
|
You have made changes according to my comments, but unfortunately, not all of them as you claimed in your correspondence. One of them is the spelling of dietician can be seen in several places (lines 16, 18, 23, 25, 26, 27, 29, 106, 108, 109, 110, 112, 121, 133, 135, 190, 239, 240, and 246). In fact, you have changed all “dietitian” into “dietician”. |
There has been a misunderstanding on our side concerning the prefered spelling of the word. We have now changed dietician to dietitian. We do hop that this is what the referee did mean. |
|
I am also concerned about the conclusion in the abstract section as the second sentence contradicts the first sentence. |
Thank you for your feedback. We understand that the first and second sentences could potentially be interpreted as contradictory. The second sentence is intended to place the initial observation within a broader context: while our survey demonstrates some progress, we aimed to emphasize that substantial improvements are still hindered by persistent barriers.
To address this, we propose the following clarification: The increasing availability of dietitians and the use of screening tools in pediatric departments suggest encouraging but limited improvement in nutritional awareness in Belgium. Persistent barriers that have remained unchanged since 2014 continue to hinder substantial advancements in nutritional care. |
|
It is strange to me that In Figure 1, the p-value for organization/follow-up screening is 0.08, while it is 0.005 for follow-up intake, even with a similar distribution. Make sure the statistics are correct. |
Thank you for bringing this to our attention. You are correct, This was a typo. The correct p-value for organization/follow-up screening is 0.008, and for follow-up intake, it is 0.005. We have thoroughly reviewed all other figures and the text against our SPSS output and can confirm that no additional typos were found. |
|
It is well described in Figure 1 that small centers (<20 beds) and large centers (> 20 beds), so the same message does not need to be reiterated in Figures 3 and 4. |
Was deleted. |
|
Do you mean “asked” instead of “informed” in line 146? |
We changed to "asked". |
|
This is a follow-up study after your 2016 publication. I would like to know whether your pediatric society/organization launched a national initiative to advocate for the implementation of your proposed guidelines. Publishing a paper cannot guarantee that all readers will accept the conclusion unless there is an authoritative instruction. |
Th Belgian Society of Paediatrics did not undertake an "official action", but our group published a paper to raise awareness (ref 7 in this manuscript). |
|
I still have some difficulty understanding some of the sentences due to word choice or the way of expression |
The manuscript was checked by a native English speaking colleague. |
|
|
|